# Pharmacological Utilization of Bergamottin, Derived from Grapefruits, in Cancer Prevention and Therapy

**DOI:** 10.3390/ijms19124048

**Published:** 2018-12-14

**Authors:** Jeong-Hyeon Ko, Frank Arfuso, Gautam Sethi, Kwang Seok Ahn

**Affiliations:** 1Department of Science in Korean Medicine, Kyung Hee University, 24 Kyungheedae-ro, Dongdaemun-gu, Seoul 02447, Korea; gokjh1647@gmail.com; 2Comorbidity Research Institute, College of Korean Medicine, Kyung Hee University, 24 Kyungheedae-ro, Dongdaemun-gu, Seoul 02447, Korea; 3Stem Cell and Cancer Biology Laboratory, School of Pharmacy and Biomedical Sciences, Curtin Health Innovation Research Institute, Curtin University, Perth 6009, Australia; frank.arfuso@curtin.edu.au; 4Department of Pharmacology, Yong Loo Lin School of Medicine, National University of Singapore, Singapore 117600, Singapore

**Keywords:** bergamottin, cancer, chemoprevention, phytochemicals

## Abstract

Cancer still remains one of the leading causes of death worldwide. In spite of significant advances in treatment options and the advent of novel targeted therapies, there still remains an unmet need for the identification of novel pharmacological agents for cancer therapy. This has led to several studies evaluating the possible application of natural agents found in vegetables, fruits, or plant-derived products that may be useful for cancer treatment. Bergamottin is a furanocoumarin derived from grapefruits and is also a well-known cytochrome P450 inhibitor. Recent studies have demonstrated potent anti-oxidative, anti-inflammatory, and anti-cancer properties of grapefruit furanocoumarin both in vitro and in vivo. The present review focuses on the potential anti-neoplastic effects of bergamottin in different tumor models and briefly describes the molecular targets affected by this agent.

## 1. Introduction

There has been considerable interest in the use of dietary compounds for various cancer prevention and therapy approaches [1,2,3,4,5,6,7,8,9,10,11,12,13,14,15,16]. Furanocoumarins are natural plant constituents present in many types of plants belonging to the Rutaceae and Umbelliferae families. Generally, furanocoumarins are primarily known to act as plants’ defense mechanism against predators and are regarded as natural pesticides [17,18]. Bergamottin is a major furanocoumarin and a bioactive component of grapefruits (*Citrus paradisi*) and other citrus fruits [19]. It was originally found in the oil of bergamot (*Citrus bergamia*), from which its name has been derived [20]. It acts as an inhibitor of some isoforms of the cytochrome P450 (CYP) enzyme, particularly CYP3A4 [21,22]. Bergamottin is also able to suppress the activities of CYP1A2, 2A6, 2C9, 2C19, 2D6, 2E1, and 3A4 in human liver microsomes [21]. For this reason, it has been recommended that patients should preferably avoid the consumption of grapefruit or grapefruit juice when they are taking prescribed medications such as statins, antihistamines, and several other orally administered drugs. The consumption of a single 6-oz glass of grapefruit juice can cause the maximal effect with enhanced bioavailability observed up to 24 h after the administration [23]. These drug interactions are often referred to as the “grapefruit effect” and can lead to increased concentrations of the affected drugs in the bloodstream, which increases the risk of potentially serious side effects from the drugs [20,24,25,26,27]. Bergamottin and the chemically related compound 6′,7′-dihydroxybergamottin are found to be responsible for this effect [20]. However, recent studies have also explored the potential benefits of CYP enzyme inhibition [28], and thus bergamottin may also be developed as an agent that can be targeted to increase the oral bioavailability of other pharmacological drugs [29]. Many studies have also demonstrated that grapefruit juice can augment the bioavailability of drugs that are CYP3A4 substrates [20,30], whereas no significant alterations were found for some other drugs [31,32]. It can exhibit a variety of interactions with drugs, leading to a reduction in therapeutic efficacy and to an augmentation of adverse effects at the same time. A variety of mechanisms, including the involvement of P-glycoprotein present in intestinal epithelium, have been proposed to explain the possible interactions of grapefruit juice with different drugs [33]. Moreover, at high grapefruit juice concentrations, P-glycoprotein-regulated vinblastine efflux was inhibited [34], whereas at low concentrations, the pumping of P-glycoprotein substrates was enhanced [35,36]. However, among various reported interactions of drugs with grapefruit juice, only a few are clinically relevant, whereas others studies predominantly involve the use of large quantities of the juice, which can be easily avoided in real-life situations to prevent the harmful effects of such interactions [37].

Additionally, there are few reports about the possible interactions of anti-cancer agents with grapefruit juice [38], and these are briefly summarized in Table 1. For example, a study analyzing the interaction of grapefruit juice with etoposide in six patients reported an unexpected decrease of 26.2% in the area under the concentration–time curve (AUC) after oral treatment [39]. Another article, which evaluated the effect of the administration of grapefruit juice with nilotinib in 21 healthy individuals, reported a 60% increase in the maximum serum concentrations (*C*_max_) and a 29% increase in the AUC without a significant effect on the half-life. Moreover, no adverse effects were noted in this study [40]. In another study, the interaction of grapefruit juice with sunitinib was analyzed in eight patients, and its co-administration increased the bioavailability of sunitinib by 11% without any increase in toxicity [41]. Interestingly, Cohen and coworkers also analyzed the toxicity and pharmacokinetic profile of intermittently administered sirolimus in patients with advanced malignancies when sirolimus was co-administered with two different CYP3A inhibitors, including grapefruit juice. They found that the grapefruit juice increased the sirolimus AUC by approximately 350%, although different grapefruit formulations may differ in their interaction profile with prescription drugs depending upon the content of various furanocoumarins present in them [42]. On the contrary, Schubert et al. reported that grapefruit juice can increase the 48-hour AUC of estradiol (E_2_) by approximately 40% after a single oral dose of E_2_ in ovariectomized women [43]. Weber et al. elaborated that grapefruit juice increased the *C*_max_ of ethinylestradiol by 38% and the 24-hour AUC by 28% [44]. Furthermore, Monroe and coworkers analyzed in a multiethnic cohort study of 46,080 postmenopausal women with 1657 cases of breast cancer whether grapefruit consumption was associated with an increased risk of breast cancer. They found that the risk was 30% higher in women who consumed the equivalent of one quarter of a fresh grapefruit or more per day, although the potential effects of diverse interactions between long-term grapefruit consumption and serum hormone concentrations still remain unclear [45]. Overall, additional studies are needed to investigate the potential interactions of bergamottin with anti-cancer agents.

Overall, the furanocoumarins can exhibit several pharmacological properties, including those of antioxidant, anti-inflammatory, and anti-cancer activities [19]. Recently, intensive interest has focused on the chemopreventive and anti-cancer potential of bergamottin. Bergamottin has demonstrated significant anti-cancer activity in skin, myeloma, leukemia, lung cancer, and other cancer cells. The present review illustrates the role of bergamottin in chemoprevention and its potential for cancer prevention and therapy.

## 2. Chemical Properties of Bergamottin

Furanocoumarins consists of a furan ring fused with coumarin and subdivided into the linear or psoralen type and the angular or angelicin type. In the linear furanocoumarins, the furan ring is connected to the benzopyrone in the carbon 6 and 7 positions, whereas the angular furanocoumarins have it fused in the carbon 7 and 8 positions (Figure 1). Its elementary composition is C_21_H_22_O_4_ and its molecular weight is 338.4 g/mol. 

Umbelliferone is often regarded as the parent of the more complex furanocoumarins, both structurally and biogenetically. The biosynthesis of bergamottin starts with the formation of demethylsuberosin, which is formed via the alkylation of umbelliferone [46]. Demethylsuberosin is transformed into marmesin by the CYP monooxygenase catalyst in the presence of NADPH and oxygen [47]. This process is then repeated to remove the hydroxyisopropyl substituent from marmesin to form psoralen and then to add a hydroxyl group at the 5-position to form bergaptol [48]. Bergaptol is next methylated with S-adenosyl methionine to form bergapten. The final step in this biosynthesis is the attachment of a geranyl pyrophosphate to the newly methylated bergapten to generate bergamottin (Figure 2).

## 3. Metabolism of Bergamottin

There are various prior reports that have highlighted the metabolic profile of furocoumarins [49]. In a pharmacokinetic study in humans, the *C*_max_ values after the administration of 6 and 12 mg bergamottin were 2.1 and 5.9 ng/mL, respectively, and the times of peak concentrations (*T*_max_) were 0.8 and 1.1 h, respectively. Interestingly, 6′,7′-dihydroxybergamottin has been detected in the plasma of some individuals after exposure to bergamottin [50]. In a study to determine the concentrations of furanocoumarins in healthy young adults before and after the ingestion of grapefruit or grapefruit juice, bergamottin and 6′,7′-dihydroxybergamottin were predominant compounds found in grapefruit flesh, juice, and plasma, while bergaptol and 6′,7′-dihydroxybergamottin were major compounds detected in the urine [51,52]. 

It has been demonstrated that the metabolism of both bergamottin and the furan ring of the psoralen moiety by CYPs can result in the formation of reactive intermediates, thereby causing inhibition of the P450 enzyme, while the metabolism of the geranyloxy chain can produce stable metabolites [48]. The metabolism of bergamottin by CYP3A4, CYP3A5, and CYP2B6 has been investigated [48,53]. Interestingly, it was found that CYP2B6 metabolized bergamottin primarily to 5′-OH-bergamottin, 6′-OH-bergamottin, and 7′-OH-bergamottin as well as to one minor metabolite (bergaptol). Because 6′- and 7′-OH-bergamottin were the primary metabolites, it was suggested that CYP2B6 can preferentially oxidize the geranyloxy chain of bergamottin. The CYP3A5 metabolism of bergamottin can also generate three major metabolites, i.e., bergaptol, 5′-OH-bergamottin, and 2′-OH-bergamottin, as well as two minor metabolites, i.e., 6′,7′-dihydroxybergamottin and 6′- and 7′-OH-bergamottin, whereas CYP3A5 and CYP2B6 induced the formation of bergamottin metabolites that can form active glutathione conjugates [48]. 

## 4. Bergamottin and Cancer

Exposure to furanocoumarins in large doses combined with ultraviolet radiation, such as through photochemotherapy, is known to induce skin tumorigenesis in both animals and humans. Recent epidemiological data suggest that relatively high levels of dietary exposure to furanocoumarins may also increase the risk of skin cancer [49]. In particular, psoralen, 5-methoxypsoralen (5-MOP), and 8-methoxypsoralen (8-MOP) are well known for their phototoxic, photomutagenic, and photocarcinogenic properties [54,55]. Recently, it has been shown that bergamottin does not exert any significant photomutagenicity on its own, as tested by a model of photomutagenicity of some furanocoumarins in V79 cells using 5-MOP as a reference compound [56]. Interestingly, the potential protective effects of furanocoumarins have also been studied in various cancer models. Table 2 briefly summarizes the potential effects of bergamottin against several cancer types and summarizes the biological mechanisms underlying its anti-neoplastic actions. 

### 4.1. Multiple Myeloma

Our group has investigated the anti-cancer potential of bergamottin in multiple myeloma (MM) cells [57]. In this study, bergamottin inhibited proliferation and induced apoptosis in human U266 MM cells through the downregulation of the signal transducer and activator of transcription 3 (STAT3) signaling pathway, which has been closely associated with tumorigenesis [70,71,72,73,74,75,76,77,78,79,80]. This suppression was mediated through the inhibition of phosphorylation of Janus-activated kinases (JAK) 1 and 2 and c-Src, as well as the induction of tyrosine phosphatase SHP-1. Furthermore, bergamottin caused a substantial down-modulation of the expression of various oncogenic proteins and significantly promoted the apoptotic effects of bortezomib and thalidomide, two drugs commonly used to treat MM. [57].

### 4.2. Leukemia

The anti-proliferative activity of bergamottin against promyelocytic leukemia HL-60 cells has also previously been reported by our group [58,59]. We observed that the combination of bergamottin and simvastatin produced synergistic effects on the tumor necrosis factor (TNF)-induced cytotoxicity and apoptosis in human chronic myelogenous leukemia KBM-5 cells. The anti-proliferative and pro-apoptotic effects of this combination therapy were found to be mediated through the suppression of the nuclear factor kappa-light-chain-enhancer of activated B cells (NF-κB), a master transcription factor regulating tumor growth and survival [60,81,82,83,84,85,86,87,88,89,90,91,92]. 

### 4.3. Skin Cancer

Polycyclic aromatic hydrocarbons (PAHs) such as benzo[α]pyrene (B[α]P) and 7,12-dimethylbenz[α]anthracene (DMBA) are routinely employed to initiate skin cancer in mouse models [93]. Bergamottin has been reported to reduce the formation of water-soluble metabolites of B[α]P and to abrogate the binding of B[α]P to DNA. It also abrogated the formation of DNA adducts derived from the anti-diol-epoxide diastereomers from both B[α]P and DMBA [61]. In another study, bergamottin was analyzed for its potential effects on the formation of B[α]P and DMBA DNA adducts in mouse epidermis. Moreover, bergamottin was noted to significantly decrease the covalent binding of B[α]P to DNA in a dose-dependent fashion, but did not significantly affect the covalent binding of DMBA to epidermal DNA at two different concentrations [69].

Interestingly, Kleiner et al. reported that bergamottin can suppress the metabolism of DMBA to DMBA-3,4-diol and block DNA adduct formation in mouse hepatoma-derived 1c1c7 (Hepa-1) cells but had a relatively minimal effect in mouse embryo fibroblast C3H/10T1/2 (10T1/2) cells. The findings of this study also indicated that bergamottin can function as a more selective inhibitor of P450 1a1 but appeared to be less potent in blocking the metabolic activation of DMBA in mouse epidermis [94]. In another study by the same group, it was found that although bergamottin was not effective at blocking DMBA bioactivation in the mouse skin model, it could abrogate the bioactivation of both DMBA as well as B[α]P in breast cancer MCF-7 cells [66].

### 4.4. Lung Cancer

The anti-cancer properties of bergamottin were also evaluated in human non-small cell lung carcinoma A549 cells. The anti-cancer effects of bergamottin were linked to an inhibited activity of colony formation, cell invasion, and cell migration in A549 cells. Furthermore, bergamottin induced apoptosis and cell cycle arrest at the G2/M phase, and it caused a significant reduction in the mitochondrial membrane potential [62]. In the mouse xenograft model of A549 cells, bergamottin showed a significant decrease of the tumor volume and weight after 18 days of consecutive treatment [62]. In a recent study from our group, bergamottin was shown to exhibit an inhibitory effect on the epithelial-to-mesenchymal transition (EMT) process in lung cancer cells [63]. EMT can facilitate the transition from a sessile epithelial state to a motile, invasive mesenchymal state and thereby cause the tumor cells to undergo metastasis to distant sites [95,96]. Interestingly, bergamottin was found to suppress transforming growth factor beta (TGF-β)-induced EMT and the cell invasive potential. This effect was found to be mediated by its inhibitory effect on PI3K, Akt, and mTOR kinases [63].

### 4.5. Fibrosarcoma

The inhibitory effects of bergamottin on metastasis and its possible mechanisms of action were also investigated in human fibrosarcoma HT-1080 cells [64]. Matrix metalloproteinases (MMPs) are actively involved in the metastasis of cancer cells, and the drug was found to substantially reduce the phorbol 12-myristate 13-acetate (PMA)-induced activation of MMP-9 and MMP-2 and to inhibit cell invasion and migration. Its anti-metastatic effects were mediated via the downregulation of NF-κB activation and the phosphorylation of p38 mitogen-activated protein kinase and c-Jun N-terminal kinase. 

### 4.6. Other Cancers

The anti-proliferative activity of bergamottin against human liver cancer HepG2 cells and gastric cancer BGC-823 cells has also been reported [59]. The cytotoxic effect of bergamottin on gastric cancer NCI-N87 cells has also been reported [65]. Additionally, studies have indicated that citrus fruit intake may reduce the risk of gastric cancer [97,98,99]. Bergamottin inhibited the proliferation of human breast cancer MDA-MB-231 and prostate cancer DU145 cells [57]. In human neuroblastoma SH-SY5Y cells, bergamot essential oils (BEOs) were also found to exhibit significant anti-proliferative effects [67] and it was hypothesized that bergamottin and 5-geranyloxy-7-methoxycoumarin may have substantially contributed to the BEO-induced anti-proliferative effects. Bergamottin also exhibited anti-invasive activity in human glioma cells through the inactivation of Rac1 activity and the downregulation of MMP-9 [68].

In summary, several studies using animal models and different cancer cell lines provide substantial evidence that bergamottin has beneficial effects against a variety of cancers. These effects are mostly attributed to its ability to regulate several cancer-related pathways including chemical detoxification, cell cycle arrest, apoptosis, migration, invasion, and angiogenesis. Figure 3 provides a concise summary of the anti-cancer effects of bergamottin with possible underlying molecular mechanisms. Bergamottin appears to be a promising natural agent for cancer prevention and therapy, and its evaluation in human clinical trials is needed to investigate its possible anti-cancer applications either as a therapeutic agent or as adjuvant therapy. 

## 5. Conclusions

The anti-cancer activity of bergamottin has been reported against many types of cancers, as briefly summarized in this review. Bergamottin may be a suitable candidate for the development of novel agents for cancer prevention and therapy. Further studies should be undertaken to examine the pharmacokinetics, ideal dosage, long-term safety, and adverse effects of bergamottin. 

## Figures and Tables

**Figure 1 ijms-19-04048-f001:**
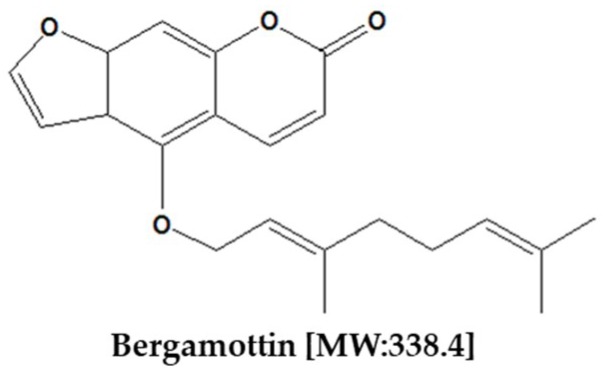
The chemical structure of bergamottin.

**Figure 2 ijms-19-04048-f002:**
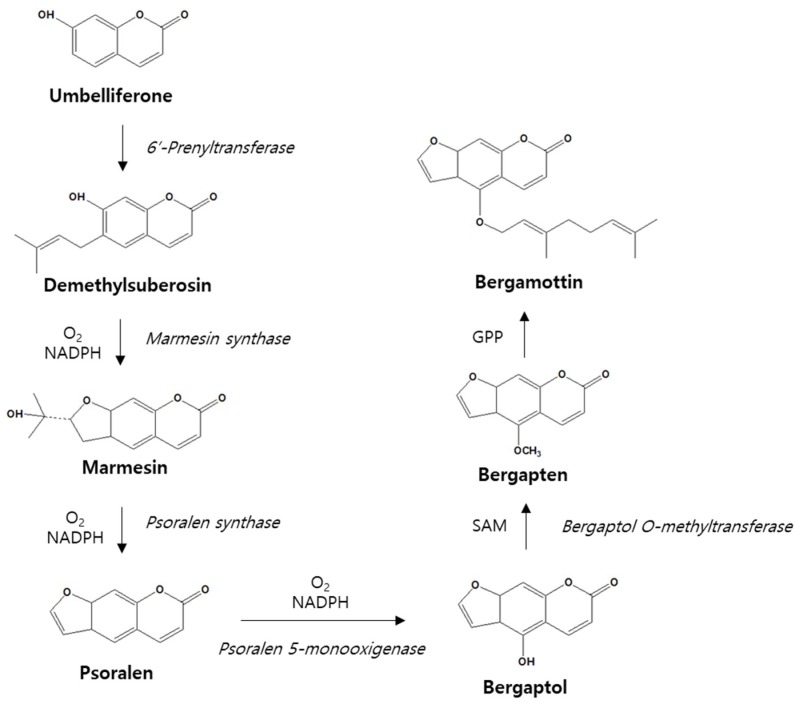
Biosynthesis pathway of bergamottin.

**Figure 3 ijms-19-04048-f003:**
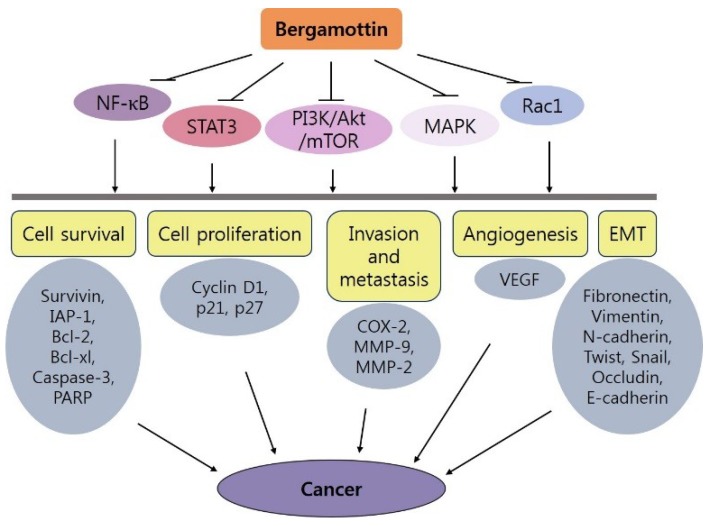
Proposed mechanisms of the anti-cancer activity of bergamottin.

**Table 1 ijms-19-04048-t001:** Reported interactions between grapefruit juice and selected anti-neoplastic drugs.

Antineoplastic Drug	Metabolism	Interaction	Ref.
Etoposide	Metabolized by CYP3A4	Decrease etoposide exposure (area under the concentration time curve (AUC) 26.2% ↓)	[39]
Nilotinib	Metabolized by CYP3A4	Increase nilotinib exposure (*C*_max_ 60% ↑, AUC 29% ↑) No increase in adverse events	[40]
Sunitinib	Metabolized by CYP3A4	Increase sunitinib exposure (*C*_max_ 10.9% ↑, AUC 11% ↑) No increase in toxicity	[41]

**Table 2 ijms-19-04048-t002:** In vitro and in vivo effects of bergamottin against malignancies.

**Type of Cancers**	**Cell Lines**	**Dose**	**Biological Effect**	**Ref.**
**In Vitro**				
Multiple myeloma	U266	50 and 100 µM for 24 h	Inhibits cell proliferation, induces apoptosis, and inhibits JAK/STAT3 activation	[57]
Leukemia	HL-60	40 µM for 4 days 6.25, 12.5, 25, and 50 µg/mL for 3 days	Inhibits cell proliferation	[58] [59]
	KBM-5	50 µM for 12 h in combination with 10 µM simvastatin	Combination with simvastatin exhibits synergistic effects of TNF-induced cytotoxicity and apoptosis	[60]
Skin cancer	Mouse epidermal keratinocytes	2 nM	Inhibits DNA adduct formation induced by B[α]P) and DMBA	[61]
Lung cancer	A549	10, 25, and 50 µM for 48 h	Induces apoptosis, cell cycle arrest, and loss of mitochondrial membrane potential Inhibits cell migration and invasion	[62]
	A549	100 µM for 24 h	Suppresses EMT, TGF-β-induced EMT, and cell invasive potential	[63]
Fibrosarcoma	HT-1080	5, 25, and 50 µM for 24 h	Reduces PMA-induced MMP-9 and MMP-2 activation Inhibits cell invasion and migration	[64]
Liver cancer	HepG2	6.25, 12.5, 25, and 50 µg/mL for 3 days	Abrogates cell proliferation	[59]
Gastric cancer	BGC-823	6.25, 12.5, 25, and 50 µg/mL for 3 days	Inhibits cell proliferation	[59]
	NCI-N87	4, 20, and 100 µM for 48 h	Attenuates cell proliferation	[65]
Breast cancer	MDA-MB-231	100 µM for 6 h 100 µM for 75 h 100 µM for 24 h	Inhibits STAT3 activation Suppresses cell proliferation Attenuates cell invasion	[57]
	MCF-7	40 µM for 24 h	Inhibits DNA adduct formation induced by B[α]P and DMBA	[66]
Prostate cancer	DU145	100 µM for 6 h 100 µM for 75 h 100 µM for 24 h	Suppresses STAT3 activation Inhibits cell proliferation Inhibits cell invasion	[57]
Neuroblastoma	SH-SY5Y	BEO (0.01, 0.02, and 0.03%) for 24 h	Suppresses cell proliferation	[67]
Glioma	U87, U251	2 and 10 µM for 48 h	Exhibits anti-invasive activity through the inactivation of Rac1 and the downregulation of MMP-9	[68]
**In Vivo**				
**Type of Cancers**	**Animal Models**	**Dose**	**Biological Effect**	**Ref.**
Skin cancer	SENCAR mice (B[α]P)	400 nmol; 5 min pretreatment	Suppresses B[α]P-induced tumor initiation	[69]
Lung cancer	BALB/c nude mice xenograft model (A549)	25, 50, and 100 mg/kg; daily; 18 days	Inhibits lung cancer growth	[62]

Abbreviations: B[α]P: Benzo[α]pyrene; DMBA: 7,12-Dimethylbenz[a]anthracene; JAK/STAT3: Janus-activated kinases/Signal transducer and activator of transcription 3; TNF: Tumor necrosis factor; EMT: Epithelial-to-mesenchymal transition; TGF: Transforming growth factor; PMA: Phorbol 12-myristate 13-acetate; MMP: Matrix metalloproteinase.

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
