# Peer review of "Pharmacological Utilization of Bergamottin, Derived from Grapefruits, in Cancer Prevention and Therapy"

_ijms, 2018, doi:10.3390/ijms19124048_

Reviewer 1 Report

Jeong-Hyeon Ko et al. summarized the anti-cancer effects of bergamottin against many types of cancer.  The authors could add more experimental details for each study cited in Table 1, such as dosage and length of treatment.  Also the authors could discuss more in vivo studies if there are any.  The author could discuss the studies using whole grapefruits in comparison with bergamottin if there are any. Overall, this manuscript reviewed adequate studies and is well-written. 

Author Response

We appreciate that the reviewer’s comments. As suggested, we added more information about dose and duration of treatment and added one more in vivo study in Table 1. There is no study using whole grapefruit in comparison with bergamottin.

Reviewer 2 Report

This is an interesting review manuscript about the anticancer potential of the natural coumarin bergamottin. Easily available natural products can feature suitable supplements for common cancer therapies. I recommend acceptance after minor revision:

Keywords: please correct "phtochemicals".

The authors mentioned the cancerogenic effects of furanocoumarins. Please discuss or mention briefly if bergamottin itself has documented mutagenic or cancerogenic potential.

Section "Bergamottin and cancer", Table 1: please reorder this section and table 1. For example, put the hematological malignancies first/on the top and then solid tumor diseases below.

Author Response

1. Keywords: please correct "phtochemicals".

Response: We appreciate that the reviewer’s comments. We have corrected it now.

2. The authors mentioned the cancerogenic effects of furanocoumarins. Please discuss or mention briefly if bergamottin itself has documented mutagenic or cancerogenic potential.

Response: As suggested, we have discussed it in section “Bergamottin and cancer”.

3. Section ´´Bergamottin and cancer´´, Table 1: please reorder this section and table 1. For example, put the hematological malignancies first/on the top and then solid tumor diseases below.

Response: As suggested, we have rearranged it under section ´´Bergamottin and cancer´´.

Reviewer 3 Report

This review manuscript is well organized and can be published in its current form. The paper summarizes the preventive  activity of bergamottin in the distinct caner types and covers most of the published papers in this aria. Only 3 review papers can be found in pubmed that discuss bergamottin and cancer and in fact only one is dedicated for this topic directly. This review is unique as it discusses the mechanism of action of the anti-neoplastic activity of bergamottin.

Author Response

Response: We appreciate that the reviewer’s comments.

Reviewer 4 Report

The publication is an important collection of nformation on the risks of cancer. The authors describe the possibilities of clinical use of bargamottin in cancer therapy. This publication is an important for scientific reasons and has a clinical aspect.

Author Response

(The authors gave the same response as above.)
